# Ubiquitin Proteasome System Role in Diabetes-Induced Cardiomyopathy

**DOI:** 10.3390/ijms242015376

**Published:** 2023-10-19

**Authors:** Ortal Nahum-Ankonina, Efrat Kurtzwald-Josefson, Aaron Ciechanover, Maayan Waldman, Orna Shwartz-Rohaker, Edith Hochhauser, Sam J. Meyer, Dan Aravot, Moshe Phillip, Yaron D. Barac

**Affiliations:** 1The Division of Cardiovascular and Thoracic Surgery, Rabin Medical Center, Petach-Tikva 4941492, Israel; ortalnahum@gmail.com (O.N.-A.); efratjo@clalit.org.il (E.K.-J.); maayanw@gmail.com (M.W.); orna.rohaker@gmail.com (O.S.-R.); aditho@clalit.org.il (E.H.); j.sammeyer@gmail.com (S.J.M.); aravot_dan@clalit.org.il (D.A.); 2Sackler Faculty of Medicine, Tel Aviv University, Tel Aviv 6997801, Israel; mosheph@clalit.org.il; 3The Ruth & Bruce Rappaport Faculty of Medicine, Technion—Israel Institute of Technology, Haifa 3109601, Israel; aaroncie@technion.ac.il; 4The Division of Endocrinology, Schneider Medical Center, Petach-Tikva 4920235, Israel

**Keywords:** UPS, T2DM, heart

## Abstract

This study investigated modifications to the ubiquitin proteasome system (UPS) in a mouse model of type 2 diabetes mellitus (T2DM) and their relationship to heart complications. *db/db* mice heart tissues were compared with *WT* mice tissues using RNA sequencing, qRT-PCR, and protein analysis to identify cardiac UPS modifications associated with diabetes. The findings unveiled a distinctive gene profile in the hearts of *db/db* mice with decreased levels of *nppb* mRNA and increased levels of *Myh7*, indicating potential cardiac dysfunction. The mRNA levels of *USP18* (deubiquitinating enzyme), *PSMB8*, and *PSMB9* (proteasome β-subunits) were down-regulated in *db/db* mice, while the mRNA levels of RNF167 (E3 ligase) were increased. Corresponding LMP2 and LMP7 proteins were down-regulated in *db/db* mice, and RNF167 was elevated in *Adult* diabetic mice. The reduced expression of LMP2 and LMP7, along with increased RNF167 expression, may contribute to the future cardiac deterioration commonly observed in diabetes. This study enhances our understanding of UPS imbalances in the hearts of diabetic mice and raises questions about the interplay between the UPS and other cellular processes, such as autophagy. Further exploration in this area could provide valuable insights into the mechanisms underlying diabetic heart complications and potential therapeutic targets.

## 1. Introduction

Type 2 diabetes mellitus (T2DM) is a disease commonly attributed to pancreatic insufficiency, inadequate insulin secretion, or increased insulin resistance, and accounts for the majority of all diabetes cases worldwide [1]. The number of people with T2DM has dramatically increased over the past decade and is expected to reach 592 million by 2035 [2]. This growing prevalence, combined with the increasingly early onset of the disease, has established T2DM as a global health concern and garnered significant attention in the literature over the past decade [3]. T2DM patients have been shown to carry a higher risk of numerous complications, including cardiovascular diseases [4], as T2DM compromises the cardiac tissue, causing cell damage, impaired calcium homeostasis, mitochondrial dysfunction, changes in muscle fibers, oxidative stress, and inflammation [5,6]. All of these cellular changes have been shown to cause heart failure and ischemic heart disease [7].

The UPS plays a central role in the regulation of many cellular processes in the body, such as cell cycle [8], cellular signaling [9] and programmed cell death [10]; consequently, deviations in UPS activity underlie the pathogenesis of various diseases [11]. Specifically, the UPS oversees cellular protein quality control by driving the degradation of misfolded and/or damaged proteins via the poly-ubiquitination of target proteins. There are three types of enzymes that promote this process: ubiquitin-activating enzymes (E1), ubiquitin-conjugating enzymes (E2), and ubiquitin-ligating enzymes (E3). Initially, E1 binds a ubiquitin molecule forming an ATP-dependent thioester linkage. The ubiquitin relocates to an E2 enzyme, which either transfers the ubiquitin to E3 directly or binds to E3 as an adaptor protein. Together, E2 and E3 attach ubiquitin to a lysine residue on the target protein [12,13] (Figure 1A). E3 ubiquitin ligases are known to define the substrate and tissue specificity of the UPS and are considered to play a significant role in maintaining protein quality control. This role is especially vital in maintaining cardiac tissue function due to its poor regenerative capacity [14]. After a cellular protein is tagged with a chain of at least four ubiquitin molecules, it is recognized and degraded by the 26S proteasome complex [13]. The barrel-shaped proteasome contains a 20S core particle and a 19S regulatory particle (RP) that binds 20S bilaterally, forming a 19S–20S–19S (30S) complex. 20S is responsible for the complex’s ATP-dependent catalytic activity and is composed of two external α rings and two internal β rings [15], with each ring comprising seven distinct subunits (α 1-7, β 1-7). In some cases, an 11S (PA28) RP replaces the 19S, forming either the 11S–20S–11S or the 19S–20S–11S hybrid (Figure 1B). These forms enable ubiquitin-independent degradation [16]. Some 11S–20S–11S complexes contain replacements of the 1, 2 and 5 β subunits with inducible counterparts, such as β1i (LMP2, encoded by *PSMB9*), β2i (MECL-1, encoded by *PSMB10*) and β5i (LMP7, encoded by *PSMB8*), respectively [17]. This transformation is triggered by IFN-γ and is referred to as the ‘immunoproteasome’.

Numerous studies have demonstrated the role of the UPS in the progression of cardiovascular diseases, including cardiac hypertrophy and heart failure [1,2,3]. Poor protein quality control (PQC) can lead to an undesirable accumulation of protein aggregates and increased levels of pro-hypertrophic and pro-apoptotic factors, which promote disease pathogenesis [4]. However, aside from its direct role in the degradation and clearance of above 90% of all misfolded, oxidized, and damaged proteins, the UPS also impacts cardiac, β-adrenergic signaling, cell excitability, conductance, and insulin gene transcription regulation [3,5]. The UPS is responsible for the degradation of the insulin receptor and the insulin receptor substrate, thereby contributing to insulin resistance and deficiency [3].

The UPS also regulates the transcription factors (TFs) that play major roles in diabetic cardiomyopathy (DCM). Peroxisome proliferator-activated receptor α (PPARα), the regulator of lipid metabolism, for example, is inhibited by MuRF1, an E3 ligase [3]. A study highlighting the role of ubiquilin1 in cardiac ubiquitination–proteasome coupling demonstrated that inadequacy may cause increased levels of myocardial ubiquitinated proteins, late-onset cardiomyopathy, and a shortened life span [6]. Studies have found that the UPS is modified in patients with diabetic nephropathy, the main cause of renal failure in diabetic patients [7]. Studies of heart disease have revealed that impaired UPS function can alone cause cardiomyopathy and heart failure [8]. Despite the compelling evidence pointing towards the important role of the UPS in cardiac function, and specifically in DCM [8], its full role has yet to be extensively researched [3].

Considering the cardiac changes that occur in the diabetic heart, we hypothesized that UPS malfunction might play a central role in early diabetes, leading to cardiac pathology. These changes may set the stage for future diabetes-induced heart failure and deterioration. Our aim was to identify such early events that might provide a better understanding of the mechanisms leading to diabetes-induced heart failure and inform the future treatment of these patients.

## 2. Results

*db/db* mice were used to explore the effects of the UPS in diabetes-induced cardiomyopathy [18,19,20]. Blood glucose levels, body weight, heart weight, and heart/body weight ratio were measured in *Young* and *Adult db/db* and *WT* mice (Figure 2). Glucose levels were higher in *Adult db/db* mice compared to *Adult WT* mice, (*p* < 0.0001), (Figure 2A). Furthermore, the body weight of both *Young* and *Adult db/db* mice was significantly higher than that of age-matched *WT* mice (Figure 2B). The heart weight was not significantly different between the study groups (Figure 2C); the heart/body weight ratio was significantly lower in *db/db* compared to WT mice, in both adult and young groups (Figure 2D).

Echocardiographic (ECHO) recordings of 16-week-old *db/db* and *WT* mice were taken (Table 1). The results identified a smaller left ventricular end systolic diameter (LVESD) in *db/db* mice compared to *WT* mice (*p* < 0.05), and a tendency towards a higher-end diastolic dimension. No other echocardiographic parameters, including the fractional shortening (FS), were significantly different between the groups.

Proteasome activity is different in *db/db* mice. Proteins extracted from the hearts of *Young* and *Adult db/db* mice and their age-matched *WT* mice were subjected to a proteasome activity assay. *Young db/db* mice exhibited reduced 20S proteasome activity compared with *Young WT* mice (Figure 3). Furthermore, while the *WT* mice exhibited progressively less activity with age, the *db/db* mice exhibited an opposite trend, with an increase in 20S activity (Figure 3).

### 2.1. RNA Sequencing Emphasizes the Differences between db/db and WT Mice

RNA sequencing was performed on the RNA samples extracted from cardiac tissues and compared with that of *WT* mice. A principal component analysis (PCA) plot representing the experiment variance was based on the gradual change in the expression of approximately 5000 genes. The PCA determined four homogenous groups as expected (Figure 4A). The variation in gene expression observed between the *Young* (8 weeks old) and *Adult* (12 weeks old) *db/db* mice points to a greater change in gene expression when the *db/db* mice transition from the pre-diabetic to the diabetic state. Differential expression (DE) analysis found vast differences in gene expression (>500 genes) between *Young db/db* and *Young WT* mice, and between *Adult db/db* and *Adult WT* mice. The exact number of modified genes, including the directional change in their expression (up- or down-regulation), is summarized in Figure 4.

The Volcano plot and the heatmap of DE genes are shown in Figure 5A–C. The number of up- or down-regulated genes in each comparison is detailed in Figure 5D.

Gene Ontology (GO) enrichment analysis. The results of the GO analysis showed that DE genes were enriched in biological processes, cytosolic components, and molecular function, as specified in Figure 6.

### 2.2. Cardiomyopathy- and UPS-Related Genes Are Differently Expressed in db/db Compared with WT Mice

The mRNA levels of cardiomyopathy- and UPS-related genes were evaluated. Cardiomyopathy-related genes such as B-type natriuretic peptide (*nppb*) [21], encoding the BNP protein, an indicator of heart failure, and myosin heavy chain 7 (*Myh7*) [22], encoding the beta (slow) heavy chain subunit of cardiac myosin, were both altered in *db/db* mice compared with *WT* mice. While *nppb* was down-regulated in diabetic mice regardless of age, differences were only significant in the comparison of *Adult* mice (*p* < 0.01) (Figure 7A). In contrast, the *Myh7* mRNA level was increased in *Adult db/db* mice (*p* = 0.05) (Figure 7B). The mRNA levels of tropomyosin alpha-3 (*Tpm3*), encoding the protein that stabilizes actin, were reduced with age in *WT* mice but increased with age in *db/db* mice (Figure 7C).

The gene expression of immunoproteasome components including the 11S α-unit and the three inducible β subunits, β5i, β1i, and β2i (encoded by *Psme1*, *PSMB8*, *PSMB9* and *PSMB10*, respectively) was significantly reduced in *Adult db/db* mice compared with *Adult WT* mice, regardless of age (Figure 8A–D).

In parallel, the mRNA levels of *RNF167*, a gene encoding an E3 ubiquitin ligase, were elevated in *Adult* diabetic mice compared with *Adult WT* mice (*p* < 0.05), (Figure 9A). The expression of *USP18*, a deubiquitinating enzyme (DUB), was significantly lower in diabetic mice compared with *WT* mice (*p* < 0.05), (Figure 9B). *Isg15*, encoding the ubiquitin-like protein USP18 substrate, was significantly down-regulated in diabetic mice compared with *WT* mice, regardless of age (*p* < 0.01), (Figure 9C). *Trim54* and *Trim63*, both encoding E3 ligases, were significantly altered. While *Trim54* was significantly up-regulated in *Adult* diabetic mice compared to *Young* diabetic mice (*p* < 0.01), (Figure 9E), *Trim63* was down-regulated in diabetic mice, regardless of age, (*p* < 0.001), (Figure 9F).

### 2.3. UPS-Related Protein Expression in db/db Mice

A trend of increasing RNF167 protein levels was detected in *Young db/db* mice, while its levels were significantly increased in *Adult db/db* mice (Figure 10A). LMP2 and LMP7 were both down-regulated in *Adult db/db* mice (Figure 10B,C).

This up-regulation relates to mechanisms impacting protein homeostasis, including autophagy and proteasome degradation. While the upregulation of immunoproteasome genes was expected, we detected a reduction in the protein levels of both LMP2 and LMP7 in *Adult* diabetic mice, suggesting the general immune susceptibility of the diabetic heart.

## 3. Discussion

This study investigated changes in UPS-related gene expression in the heart tissue of *db/db* mice at different stages of diabetes progression. The analysis revealed that genetic changes in UPS components were already detectable in *Young*, pre-diabetic mice, indicating an early alteration in UPS activity.

*Young db/db* mice are considered pre-diabetic, and they transition to fully diabetic at approximately 12 weeks of age [18]. The blood glucose levels in pre-diabetic *db/db* mice were slightly higher than in age-matched *WT* mice, with a significant value in the *Adult* group (Figure 2A). The heart/body weight ratio significantly declined with age in *WT*. This decline is likely due to the greater increase in the body and heart weight of *db/db* mice, while leaving the heart/body weight ratio nearly unchanged (Figure 2D). Differences in UPS expression in diabetic myocardial tissue had already been noted om *Young db/db* mice, exhibiting reduced 20S proteasome activity compared to *WT* (Figure 3). The reduced activity of 20S aligns with reports by Predmore and colleagues, demonstrating reduced proteasomal activity in failing hearts [23]. The 20S proteasome can remain free or bind to either 19S or 11S regulator units [24], while 11S can improve the ability of 20S to specifically locate oxidized proteins. Based on our RNA-seq analysis, the genes encoding the 19S particles were not different among the groups; however, the genes encoding the immunoproteasome and several cardiac genes expressed significant differences (Figure 4E,F). *nppb*, a circulatory hormone and a heart failure biomarker, was reduced in diabetic mice (Figure 7A); this was in line with findings reported by Zhang et al., who found that *nppb* synthesis is reduced following the onset of systemic insulin resistance in mice [21]. *Myh7* was up-regulated in *db/db* mice (Figure 7B), suggesting a change in cardiac muscle composition, with possible implications for contractility and functionality. The overexpression of *PSME1*, a gene encoding the PA28α unit of 11S, was found to have a positive effect on diabetic and hyperglycemic rat hearts [25,26]. The current study detected a reduction in *PSME1* mRNA levels in *db/db* mice (Figure 8A), while in *WT* mice, *PSME1* mRNA levels increased with age. These findings underscore the importance of *PSME1* in healthy cardiac tissue.

We observed that the mRNA levels of immunoproteasome β subunits, specifically up-regulated under stress conditions [27], increased with age in *WT* mice but not in *db/db* mice (Figure 8B–D). This suggests a reduced cellular response to oxidative stress in the diabetic heart, which can further impair protein homeostasis. E3 ubiquitin ligases hold a central role in the progression of cardiovascular pathologies [27], e.g., cardiac hypertrophy, apoptosis inhibition [28,29], and the regulation of cardiac reactive oxygen species (ROS) [30]. Consistent with earlier reports on the role of the E3 ligases in regulating cardiac apoptosis, metabolism, hypertrophy, ischemia and diabetic cardiomyopathy [31,32,33,34], we found that the mRNA levels of the E3 ligase *RNF167* were up-regulated in both *Young* and *Adult db/db* mice (Figure 9A). Additionally, RNF167 protein expression was significantly increased in 12-week-old *db/db* mice (Figure 10A). This up-regulation relates to the mechanisms impacting protein homeostasis, including autophagy and proteasome degradation. RNF167 regulates the lysosomal exocytosis [30,35,36] of the α-amino-3-hydroxy-5-methyl-4-isoxazolepropionic acid glutamate receptor (AMPAR), which is involved in synaptic transmission in neurons [37]. It also regulates the lysosomal exocytosis of N-methyl-D-aspartate receptors (NMDAR), which are present in the heart, and have been shown to impact Ca^2+^ levels [38]. Both RNF167 and Nedd4-1 (neural precursor cell-expressed developmentally down-regulated gene 4-1), another E3 ligase, regulate cell–cell communication [39] by facilitating the ubiquitination of AMPAR in mammalian neurons [40,41]. However, based on our RNA-Seq results, no alterations in Nedd4-1 were detected, further highlighting the specificity of RNF167 up-regulation in the heart of diabetic mice. I/R damage has similar mechanistic effects on NMDARs in both cardiomyocytes and neurons [38]. In the current study, RNF167 was one of four E3 ligases that exhibited an altered expression (in addition to Trim54/Trim63/Herc6; and was up-regulated. These results suggest the specific role of NMDARs in regulating the diabetic heart, along with the other unknown proteins that RNF167 regulates.

Diabetic cells are starved due to poor glucose transport [42], reduced levels of mitochondrial Ca^2+^, and the increased production of ROS, all of which trigger a reduction in ATP synthesis. During cell starvation, plasma membrane damage accumulates and impairs cellular homeostasis [35]. RNF167, a regulator of lysosome calcium signaling and localization [43], contributes to increased intracellular Ca^2+^ and facilitates lysosome movement towards the plasma membrane. This process enables the removal of the damaged plasma membrane and prevents it from accumulating in the cell. Thus, the up-regulation of RNF167 in 12-week-old *db/db* mice may represent an attempt to combat this phenomenon in the diabetic heart. Assuming that ligase families are involved in cardiac metabolism [14], it is logical to link RNF167 up-regulation in the diabetic heart with the enhanced protein tagging required for lysosomal degradation.

We detected a few changes in mitophagy-related genes in diabetic mice. For example, BCL2/adenovirus-interacting protein 3 (*BNIP3*), a mitophagy marker known to be up-regulated by the c-Jun N-terminal protein kinase (JNK) pathway in ER-stressed cells and known to affect contractility [41], was down-regulated compared with *WT* mice. *PARK2* (Parkin), which enhances mitophagy via ubiquitinating mitochondria [44], was up-regulated in *Adult* diabetic mice. No changes in any other autophagy genes, such as *ATGs*, *LC3*, or *PINK1*, were detected. The limited autophagy-related gene changes detected by RNA-seq in *diabetic* mice further support our hypothesis of a unique UPS compensatory mechanism underlying diabetic cardiomyopathy.

DUBs sort ubiquitinated proteins to authorize their entry into the proteasome, thereby removing and recycling ubiquitin [45]. The reduced expression of USP18, a DUB involved in regulating inflammation, was observed in diabetic mice (Figure 9B), suggesting impaired UPS regulation in the early prediabetes stages. Reductions in LMP2 (β1i) and LMP7 (β5i) protein levels were also detected, indicating the suboptimal functioning of the UPS in diabetic hearts. LMP7 is necessary for balanced protein homeostasis in a healthy aging heart [46]. Here, we detected a reduction in the protein levels of both LMP2 and LMP7 in *Adult* diabetic mice, suggesting the general immune susceptibility of the diabetic heart. While the up-regulation of immunoproteasome genes and proteins was expected, these findings proved incongruous with this prediction as the ratio of proteasome/immunoproteasome activity decreased in the heart with age. Inducible β subunits are up-regulated under stressful conditions, such as following viral invasion [21]. The increase is attributed to a compensatory mechanism activated in response to a rise in oxidized proteins [22]. Wang et al. detected the accumulation of polyubiquitinated proteins in LMP7^−/−^-deficient cells in response to interferon γ treatment [23]. Here, the mRNA levels of all three immunoproteasome β subunits increased with age in the cardiac tissue of WT mice (Figure 8B–D), but not in *db/db* mice. The absence of this response in cardiac *db/db* tissue in the early stages of diabetes likely results in a reduced cellular response to oxidative stress and to the further impairment of protein homeostasis. These results align with the reduction in 20S activity in *db/db* mice (Figure 3). According to research carried out by Jenkins et al. [24], malignant cells express a high immune/constitutive proteasome ratio, highlighting its potential role as a target for treatments and emphasizing the importance of an immune/constitutive balanced ratio. Thus, immunoproteasome-targeted inhibition may reduce the side effects of the continuing toxicities that harm the heart. db/db mice expressed a decreased ratio of immunoproteasome/constitutive with age, pointing to the susceptibility or imbalanced protein homeostasis within their heart. The reduction in LMP2 and LMP7 proteins in diabetic hearts points to the suboptimal functioning of the proteasome when compared with the hearts of the age-matched WT. As we did not capture the expected increase in these units, we may assume that a delayed one might occur later in the course of diabetic hearts, but this was beyond the scope of our work.

The optimal functioning of cardiac UPS components, including proteasome, immunoproteasome and E3 ligases, is essential to healthy cardiac function. The presented findings confirm a genetic UPS imbalance in *db/db* cardiac tissue. This improves our understanding of the changes to protein homeostasis in the heart that either precede or accompany diabetes development and involve a wide range of UPS players. These findings also raise questions regarding the crosstalk between the UPS and other cellular processes such as autophagy, mitophagy, and lysosome trafficking in the diabetic heart. Future research should focus on characterizing the network of UPS players and their respective roles in healthy heart function. Such data may assist in diagnosing UPS imbalances and may identify biomarkers of early-stage cardiac stress prior to a diagnosis of diabetes.

## 4. Materials and Methods

To test this hypothesis, we monitored UPS component expression in the hearts of *db/db Young* (pre-diabetic) and *Adult* (diabetic) mice using a T2DM model.

Animal model for T2DM: The animal experiments were approved by the Tel Aviv University Institutional Animal Care and Use Committee, permit number: M-13-044, 01-21-077. Two strains of mice were used: BLKS/J-Leprdb/-Leprdb/OlaHsd (*db/db*) mice (a model for type 2 diabetes) and C57BLKs/6JOlaHsd (*WT*) mice (wild-type control). Mice were acclimated for two weeks in a pathogen-free facility and provided with regular rodent chow and water ad libitum.

Animal study design: Mice were divided into two age groups: *Young* (6 weeks old) and *Adult* (12 weeks old).

Glucose measurements: The blood glucose levels were measured in mice after a 12 h fast using a Glucometer. The mice were weighed and anesthetized with 2% isoflurane inhalation and blood samples were collected from the tail vein for glucose measurement.Echocardiographic recordings: The mice were anesthetized with 2% isoflurane inhalation. Echocardiography was performed using a Vevo 2100 Imaging System with a 30 MHz linear transducer (VisualSonics Inc., Toronto, ON, Canada).

Two-dimensional (2D) guided M-mode echocardiography was used to assess heart function. The left-ventricular end-diastolic dimensions (LVEDD) and left-ventricular end-systolic chamber dimensions (LVESD) were measured. The left-ventricular fractional shortening (FS) was calculated as a measure of cardiac function.

RNA extraction from heart tissue: The RNA was extracted from heart tissue using the RNeasy Fibrous Tissue Mini Kit, (Qiagen, Hilden, Germany). The extracted RNA was quantified using a NANODROP, (Thermo Scientific, Waltham, MA, USA) and evaluated for RNA integrity using an RNA integrity number (RIN).

Genomics: RNA sequencing was performed at The Crown Genomics Institute of the Nancy and Stephen Grand Israel National Center for Personalized Medicine. Libraries were prepared using the G-INCPM mRNAseq protocol. Sequencing was performed using the Illumina HiSeq machine. Bioinformatics analysis was conducted to identify differentially expressed (DE) genes and determine relevant biological functions and pathways.

Quantitative real-time polymerase chain reaction (qPCR): The cDNA was prepared from RNA samples using the TaqMan High-capacity cDNA Reverse Transcription (RT) kit, (Applied Biosystems, Foster City, CA, USA). qPCR was performed using the Stepone Real-Time PCR cycle, (Applied Biosystems, Foster City, CA, USA). Specific primers were used for amplification, and the gene expression levels were quantified using the 2^−ΔCT^ method. A list of the primers used for qRT-PCR is summarized in Table 2.

20S Proteasome activity and protein analysis: The proteasome activity was measured using the APT280 20S proteasome activity kit (Millipore, Sigma, Burlington, MA, USA). The assay was conducted following the manufacturer’s instructions and was based on fluorophore 7-Amino-4-methylcoumarin (AMC) detection after cleavage from the Suc-Leu-Leu-Val-Tyr-AMC (LLVY-AMC) substrate. Protein lysate (30 µg) was mixed with 20 µL of substrate and incubated at 37 °C for 120 min; then, the AMC was measured [39,44]. To ensure that only the proteasomal activity was measured, samples treated with the proteasomal inhibitor lactacystin served as a negative control. Fluorescence was detected using a Synergy H1 microplate reader (Biotek, Winooski, VT, USA) at 380 nm excitatory (Ex) and 460 nm emission (Em) wavelengths. The enzymatic activity was calculated using a calibration curve of free AMC and following the manufacturer’s instructions.

Protein extraction from heart tissue and cells was performed using appropriate buffers. Western blot analysis was conducted to detect protein expression using specific antibodies. A list of the antibodies used for western blotting are summarized in Table 3.

Statistical analysis: Data are presented as mean ± standard deviation. Differences between experimental groups were analyzed using two-way analysis of variance (ANOVA) followed by a Tukey test or the Mann–Whitney non-parametric test. A *p*-value of <0.05 was considered statistically significant.

## Figures and Tables

**Figure 1 ijms-24-15376-f001:**
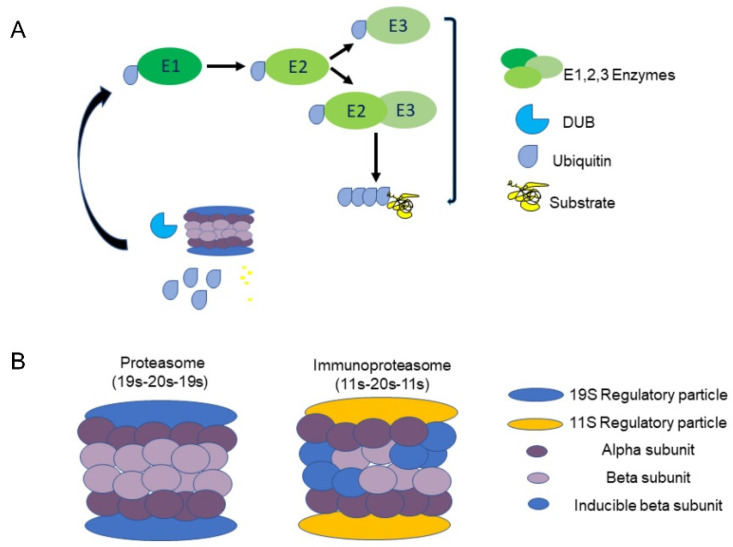
The ubiquitin proteasome system: (**A**) Three types of enzymes promote degradation: ubiquitin-activating enzymes (E1), ubiquitin-conjugating enzymes (E2), and ubiquitin-ligating enzymes (E3). Initially, E1 binds a ubiquitin molecule forming an ATP-dependent thioester linkage. The ubiquitin relocates to an E2 enzyme, which either transfers the ubiquitin to E3 directly or binds to E3 as an adaptor protein. Together, E2 and E3 attach ubiquitin to a lysine residue on the target protein. Deubiquitinating enzymes (DUBs) sort ubiquitinated proteins, and remove and recycle ubiquitin. (**B**) A constitutive proteasome 19S–20S–19S hybrid (**left**) and immunoproteasome 11S–20S–11S hybrid (**right**).

**Figure 2 ijms-24-15376-f002:**
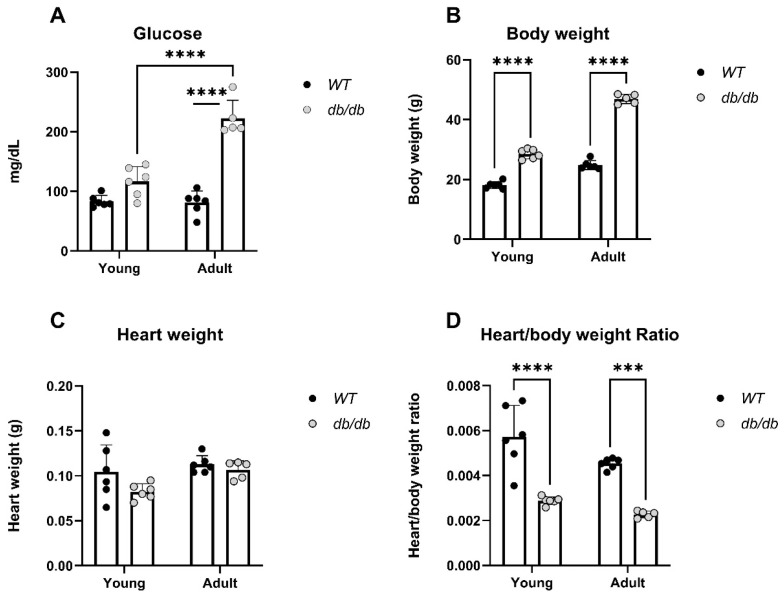
Physiological differences obtained according to glucose, body weight and heart weight measurements of *Young* and *Adult db/db* and *WT* mice. Y: 4–8-week-old mice. A: 11–18-week-old mice. Glucose: *WT*_y_: n = 6, db_y_: n = 6, *WT*_a_: n = 6, db_a_: n = 5, (**A**). Body weight: *WT*_y_: n = 6, db_y_: n = 6, *WT*_a_: n = 6, db_a_: n = 5, (**B**). Heart weight: *WT*_y_: n = 6, db_y_: n = 6, *WT*_a_: n = 6, db_a_: n = 5, (**C**). Heart/body weight: *WTy*: n = 6, *dby*: n = 6, *WTa*: n=6, *dba*: n = 5, (**D**) *** *p* < 0.001, **** *p* < 0.0001.

**Figure 3 ijms-24-15376-f003:**
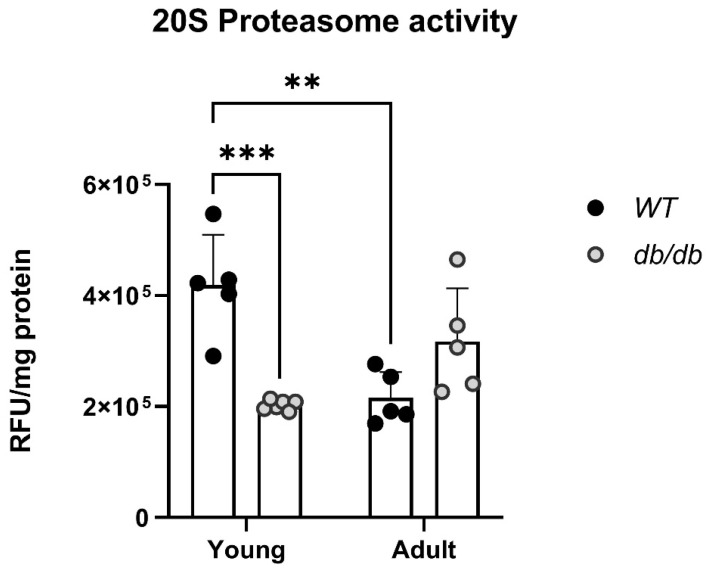
Abnormal 20S proteasome activity in diabetic mice. *Young*: 6-week-old mice (y), *Adult*: 12-week-old mice (a) *WT*_y_: n = 5, db_y_: n = 6, *WT*_a_: n = 5 db_a_: n = 5, two-way ANOVA ** *p* < 0.01, *** *p* < 0.001.

**Figure 4 ijms-24-15376-f004:**
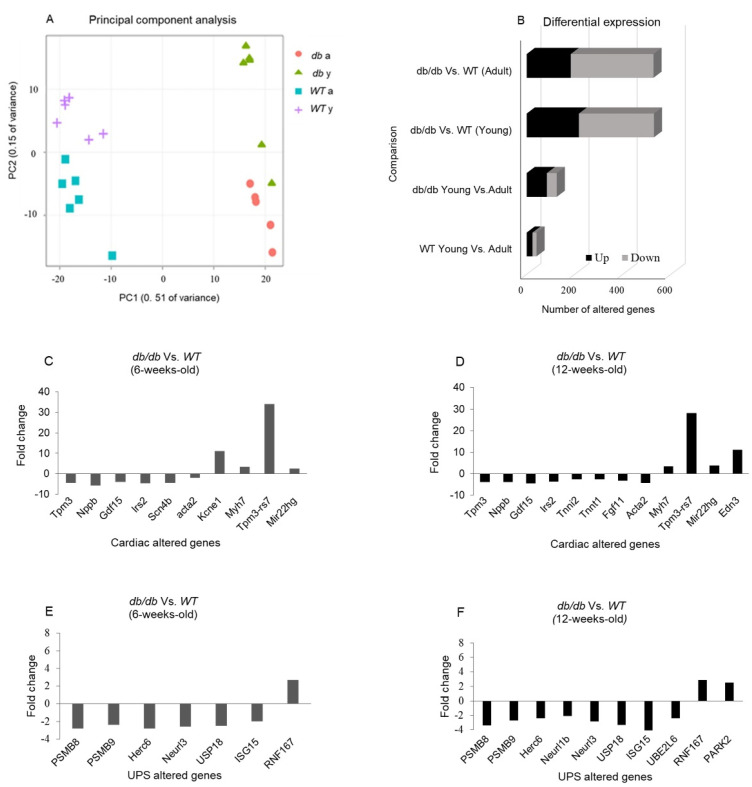
RNA sequencing of cardiac samples detected changes in over 500 genes. A principal component analysis (PCA) was performed to present the sample differences according to the gene expression levels. Threshold for significance was set at *p* adjusted ≤ 0.05, max counts per gene ≥ 30, |Fold Change| ≥ 2 (**A**). Differential expression (DE) comparisons found major gene expression differences between the age-matched *db/db* and WT mice groups (**B**). DE of cardiomyopathy- (**C**,**D**) and UPS-related genes (**E**,**F**) detected via RNA sequencing in *Young* (6-week-old) and *Adult* (12-week-old) *db/db* vs. WT mice.

**Figure 5 ijms-24-15376-f005:**
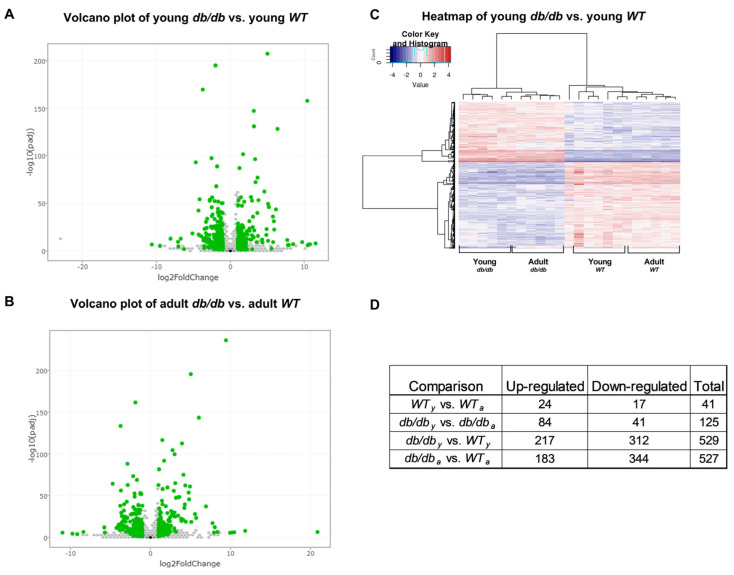
Volcano plots of differential expression: *Young db/db* vs. *Young WT* (**A**), *Adult db/db* vs. *Adult WT* (**B**), Each point on the graph represents a gene. Points having a fold change more than 2 are shown in green. Heatmap displays significantly (*p*adj ≤ 0.05 and |FC| ≥ 2) differentially expressed genes across four comparisons (*Young WT* vs. *Adult WT*; *Young db/db* vs. *Adult db/db*; *Young db/db* vs. Young *WT*; *Adult db/db* vs. *Adult WT*). Each row of the heatmap represents the z-score-transformed log2 values of one differentially expressed gene across all samples (blue, low expression; red, high expression), (**C**). Table denotes the number of genes up- or down-regulated in each comparison (**D**).

**Figure 6 ijms-24-15376-f006:**
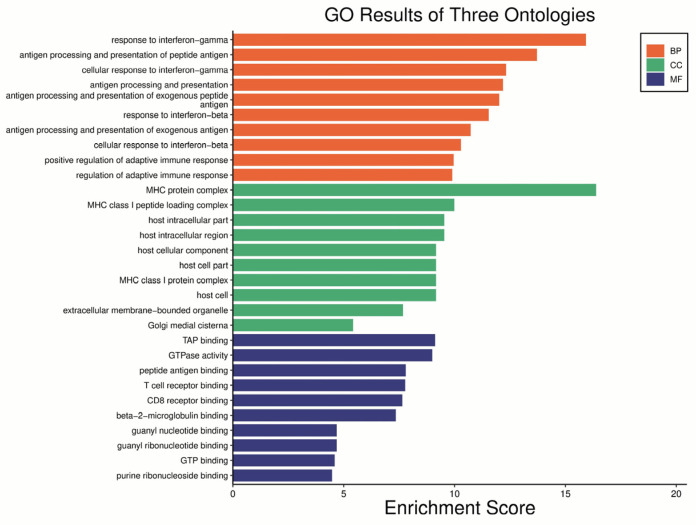
Gene ontology enrichment analysis of *Young db/db* vs. *Young WT*. BP—Biological process, CC—Cellular component, MF—Molecular function.

**Figure 7 ijms-24-15376-f007:**
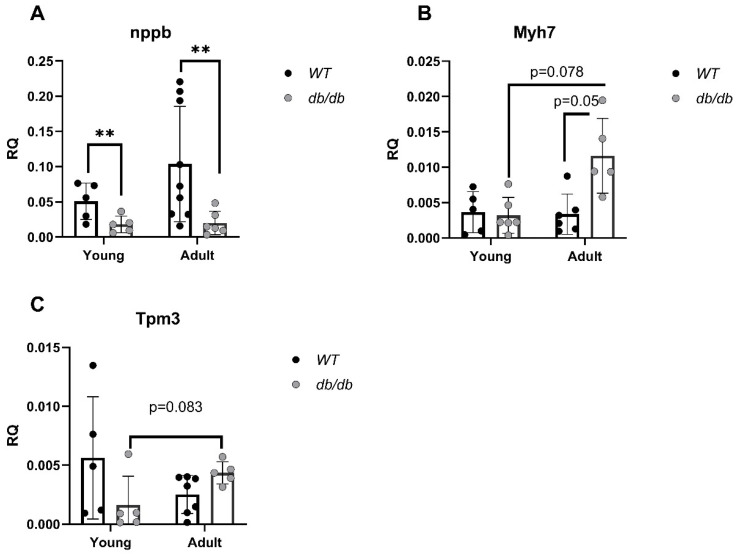
Cardiomyopathy-related gene alterations in diabetic mice, detected via qRT-PCR. *RNF167* (n = 5) (**A**), *USP18* (n = 5), (**B**), *Isg15* (n = *WT_y_*: n = 5, *db_y_*: n = 5, *WT_a_*: n = 5, *db_a_*: n = 6) (**C**), *Ube2l6* (n = *WT_y_*: n = 5, *db_y_*: n = 5, *WT_a_*: n = 5, *db_a_*: n = 6) ** *p* < 0.01 using a two-way ANOVA with interaction.

**Figure 8 ijms-24-15376-f008:**
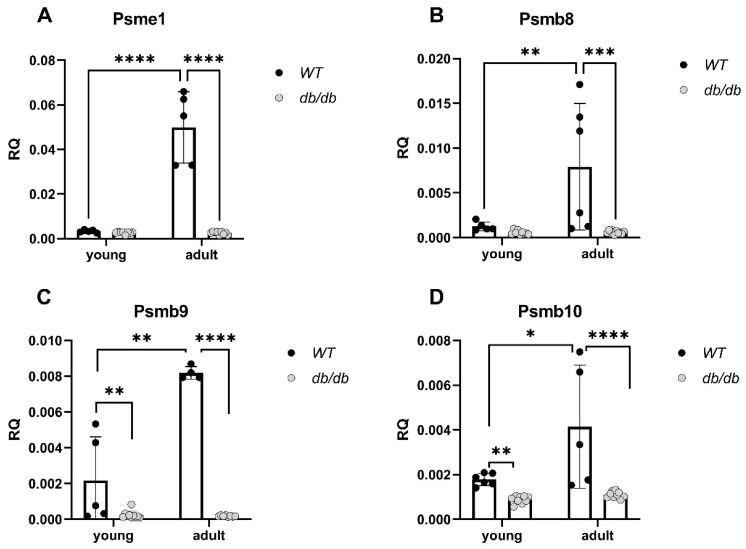
Reduced expression of genes encoding for immunoproteasome components in diabetic mice. The mRNA levels of proteasome component genes were compared between *db/db* and *WT* mice. *Psme1*: *WT*_y_: n = 5, db_y_: n = 5, *WT*_a_: n = 8, db_a_: n = 9, (**A**), *PSMB8*: *WT*_y_: n = 5, db_y_: n = 6, *WT*_a_: n = 8, db_a_: n = 9, (**B**). *PSMB9*: *WT*_y_: n = 5, db_y_: n = 4, *WT*_a_: n = 10, db_a_: n = 8, (**C**). *PSMB10*: *WT*_y_: n = 6, db_y_: n = 5, *WT*_a_: n = 9, db_a_: n = 9, (**D**). Results are normalized to GAPDH. Two-way ANOVA. * *p* < 0.05, ** *p* < 0.01, *** *p* < 0.001, **** *p* < 0.0001.

**Figure 9 ijms-24-15376-f009:**
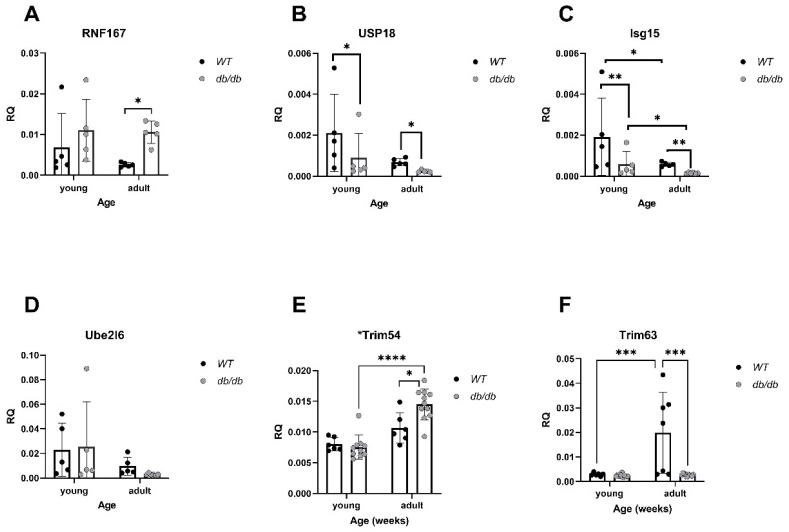
Alterations in UPS-related gene expression in diabetic mice: mRNA levels of UPS genes were compared between *db/db* and *WT* mice. *RNF167*: All groups: n = 5, (**A**). *USP18*: All groups: n = 5, (**B**). *Isg15: WT*_y_: n = 5, db_y_: n = 5, *WT*_a_: n = 5, db_a_: n = 6, (**C**). *Ube2l6*: *WT*_y_: n = 5, db_y_: n = 5, *WT*_a_: n = 5, db_a_: n = 6, (**D**), *Trim54 (Murf3)*: *WT*_y_: n = 6, db_y_: n = 6, *WT*_a_: n = 10, db_a_: n = 11, (**E**). *Trim63 (Murf1)*: *WT*_y_: n = 6, db_y_: n = 7, *WT*_a_: n = 9, db_a_: n = 10, (**F**). Results are normalized to GAPDH. Two-way ANOVA * *p* < 0.05, ** *p* < 0.01, *** *p* < 0.001, **** *p* < 0.0001.

**Figure 10 ijms-24-15376-f010:**
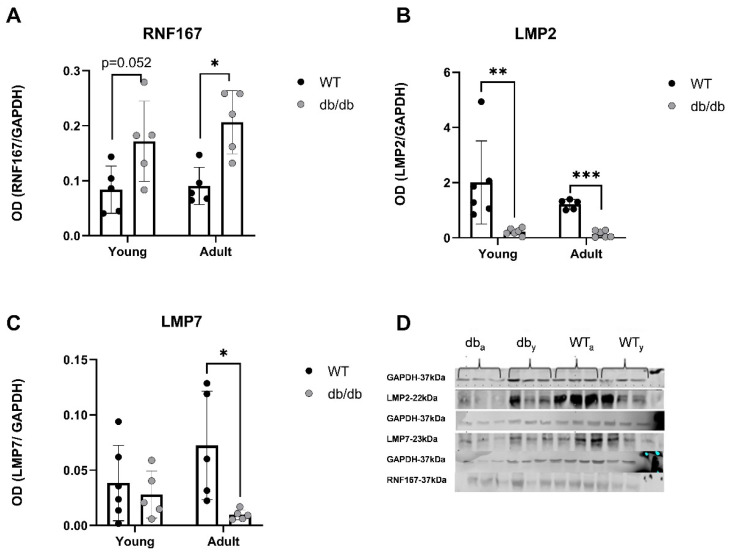
UPS-related proteins are expressed differently in the heart of *db/db* mice compared with *WT* mice. Protein expression was evaluated using western blot analysis. RNF176 (n = 5) (**A**), LMP2 (n = 5–6) (**B**) and LMP7 (n = 5–6) (**C**), *p* < 0.05) (**D**), * *p* < 0.05, ** *p* < 0.01, *** *p* < 0.001. Multiple Mann–Whitney test.

**Table 1 ijms-24-15376-t001:** Echocardiography: Interventricular septum (IVS), left ventricular posterior wall (LVPW), left ventricular end dimension diastole/systole (LVEDD/LVESD, respectively), fractional shortening (FS%). *WT* n = 7, *db*/*db* n = 11. *t*-test * *p* < 0.05.

	*WT*	*db/db*
IVS (mm)	0.8 ± 0.1	0.9 ± 0.1
LVPW (mm)	0.9 ± 0.1	0.9 ± 0.1
LVEDD(mm)	3.6 ± 0.7	3.9 ± 0.2
LVESD (mm)	2.9 ± 0.2	2.6 ± 0.3 *
FS (%)	33 ± 14	34 ± 0.7

**Table 2 ijms-24-15376-t002:** A list of the primers used for qRT-PCR.

Gene Primers	Catalog Number	Company
*GAPDH*	Mm99999915_g1	Applied Biosystems, Thermo Fisher scientific, Foster City, CA, USA
*PSME1*	Mm00650858_g1	Applied Biosystems, Thermo Fisher scientific, Foster City, CA, USA
*PSMB8*	Mm01278980_g1	Applied Biosystems, Thermo Fisher scientific, Foster City, CA, USA
*PSMB9*	Mm00479004_g1	Applied Biosystems, Thermo Fisher scientific, Foster City, CA, USA
*PSMB10*	Mm00479052_g1	Applied Biosystems, Thermo Fisher scientific, Foster City, CA, USA
*USP18*	Mm01188805_m1	Applied Biosystems, Thermo Fisher scientific, Foster City, CA, USA
*ISG15*	Mm01705338_s1	Applied Biosystems, Thermo Fisher scientific, Foster City, CA, USA
*RNF167*	Mm00550965_m1	Applied Biosystems, Thermo Fisher scientific, Foster City, CA, USA
*UBE2l6*	Mm00498295_m1	Applied Biosystems, Thermo Fisher scientific, Foster City, CA, USA
*Myh7*	Mm00600555_m1	Applied Biosystems, Thermo Fisher scientific, Foster City, CA, USA
*Nppb*	Mm01255770_g1	Applied Biosystems, Thermo Fisher scientific, Foster City, CA, USA
*Tpm3*	Mm04336671_g1	Applied Biosystems, Thermo Fisher scientific, Foster City, CA, USA

**Table 3 ijms-24-15376-t003:** List of the antibodies used for western blotting.

Primary Antibodies	Catalog Number	Company
GAPDH (1:1000)	sc-36502	Santa Cruz Biotechnology, Dallas, TX, USA
RNF167 (1:100)	ab185099	Abcam, Cambridge, UK
Lmp2 (1:100)	ab3328	Abcam, Cambridge, UK
Lmp7 (1:100)	ab3329	Abcam, Cambridge, UK
**Secondary Antibodies**	**Cat Number**	**Company**
Goat anti-rabbit IgG, IRDye®800RD (1:10,000)	925-32213	LI-COR Corporate, Lincoln, NE, USA
Donkey anti-mouse IgG, IRDye®680RD (1:10,000)	925-98072	LI-COR Corporate, Lincoln, NE, USA

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
