# Peer review of "Ubiquitin Proteasome System Role in Diabetes-Induced Cardiomyopathy"

_ijms, 2023, doi:10.3390/ijms242015376_

Round 1

Reviewer 1 Report

The authors have presented a novel manuscript describing changes in the ubiquitin-proteasome system and its probable consequences in the diabetic heart. I don’t have any significant concerns about the manuscript, but a few issues need to be addressed before accepting the manuscript.

General comments

1)    It would be beneficial to include information regarding the role of UPS in cardiac and diabetes physiology in the manuscript's introduction.

2)    The paper should include raw images of echocardiography data (M-mode). The data presented as a graph will be more informative, and the number of mice per group should be included in the figure legend.

3)    It appears that authors still need to fully utilize the RNA sequencing data. To enhance the paper's quality, they should incorporate a volcano plot illustrating the variations in gene expression between WT and dbdb mice. Additionally, it would be beneficial to include a GO analysis of upregulated and downregulated genes.

4)    It may be beneficial to consider including some rationalization and interpretation of the data in the results section for a clearer presentation.

5)    The authors should address the effects of upregulating RNF167 and the resulting downregulation of LMP2 and LMP7 in the discussion section. They should also explain why these genes are differentially regulated.

Author Response

We thank the reviewer for his remarks and the article was rewritten and corrected accordingly:

Reviewer 2 Report

The ubiquitin-proteasome system (UPS) plays a crucial role in maintaining protein homeostasis (proteostasis) within cells by regulating the degradation of unwanted or damaged proteins. It involves the tagging of proteins with ubiquitin molecules, which target them for degradation by the proteasome, a large multi-subunit protein complex. In the context of type 2 diabetes mellitus (T2DM), the UPS has been implicated in several aspects of the disease, including insulin resistance, β-cell dysfunction, and the development of diabetic complications.

While there is evidence suggesting the involvement of the UPS in T2DM, the precise mechanisms and interactions are still being elucidated. Here authors tried to address an important aspect of UPS in T2DM. However, current manuscript did not present good quality data, also largely lacks scientific depth. The introduction did not establish any link between the paragraph without any clear hypothesis. Discussion and result section, materials and methods is of very poor quality, and having no proper interpretation and analysis.

According to me the manuscript really need in-dept revision and also authors need to fully rewrite it. However, I am here suggesting certain things to consider during next submission.

1) All the figures are in poor quality.

2) Line 76-81, authors abruptly jumped into role of UPS in diabetic. Need to cite references linked to UPS malfunction and diabetic heart. The paragraph is not very coherent with the earlier section of the introduction. Need to elaborate on this further in brief.

3) Authors did not add any discuss for Figure 2C. In this figure, does the heart weight has any significant difference in young and adult WT vs db/db mice?

4) As written in Figure 2 legend, for better understanding authors may provide additional diagram, marking the young and old mice.

5) Why authors decided to check the 20s proteasome activity not the 26S proteasome? Authors must describe how the proteasome activity (Figure 3) measured and the relevant reagent details. The description in material method section is very limited. It's necessary to show that difference between db/db young and adult in 20S proteasome activity is significant.

6) Line 124, authors mentioned about >500 differentially expressed gene. Here, authors must present a GO analysis of 500 differentially expressed genes. Also, authors need to show entire heatmap of all differentially expressed genes in different condition as supplementary. In figure 4, authors must mentioned in the text and legend what are dba, dby, Wta, WTb represent for clear understanding.

7) Why authors suddenly decided to change the show the difference in the from p value than star (*). The format should be consistent. Fig 5B, 5C.

8) In this figure 6, it is presented a significant reduction in expression of genes encoding immunoproteasome component in db/db adult mice compare to WT, however in figure 3 authors observed a significant increase of 20S proteasome activity in db/db adult mice compare to WT, the reason is  not clear. Authors need to briefly discuss on this, which is just missing here.

9) Authors should be consisted with other figures. In figure 8, authors should use either p value or star (*). Should not mix. poor blot quality in figure 8D. Authors must present a decent quality of a blot.

10) In the figure 8, authors showed changes in RNF176, LMP2 and LMP7 gene. Authors must provide the differentially expressed heart in different conditions as supplementary.

Author Response

(The authors gave the same response as above.)

Round 2

Reviewer 2 Report

The revised manuscript is significantly improved and added necessary details. The authors are applauded for taking the comments seriously and committing to address them as fully as possible.